# Comparative Proteomic Analysis of Floral Buds before and after Opening in Walnut (*Juglans regia* L.)

**DOI:** 10.3390/ijms25147878

**Published:** 2024-07-18

**Authors:** Haoxian Li, Lina Chen, Ruitao Liu, Shangyin Cao, Zhenhua Lu

**Affiliations:** 1Zhengzhou Fruit Research Institute, Chinese Academy of Agricultural Sciences, Zhengzhou 450000, China; lihaoxian@caas.cn (H.L.); chenlina@caas.cn (L.C.); liuruitao@caas.cn (R.L.); s.y.cao@163.com (S.C.); 2National Horticultural Germplasm Resources Center, Chinese Academy of Agricultural Sciences, Zhengzhou 450000, China; 3National Nanfan Research Institute, Sanya 572000, China; 4Zhongyuan Research Center, Chinese Academy of Agricultural Sciences, Xinxiang 453000, China; 5National Key Laboratory for Germplasm Innovation & Utilization of Horticultural Crops, Zhengzhou 450000, China

**Keywords:** walnut (*Juglans regia* L.), proteomics, floral buds, pistil and stamen development, iTRAQ

## Abstract

The walnut (*Juglans regia* L.) is a typical and an economically important tree species for nut production with heterodichogamy. The absence of female and male flowering periods seriously affects both the pollination and fruit setting rates of walnuts, thereby affecting the yield and quality. Therefore, studying the characteristics and processes of flower bud differentiation helps in gaining a deeper understanding of the regularity of the mechanism of heterodichogamy in walnuts. In this study, a total of 3540 proteins were detected in walnut and 885 unique differentially expressed proteins (DEPs) were identified using the isobaric tags for the relative and absolute quantitation (iTRAQ)-labeling method. Among all DEPs, 12 common proteins were detected in all four of the obtained contrasts. GO and KEGG analyses of 12 common DEPs showed that their functions are distributed in the cytoplasm metabolic pathways, photosynthesis, glyoxylate and dicarboxylate metabolism, and the biosynthesis of secondary metabolites, which are involved in energy production and conversion, synthesis, and the breakdown of proteomes. In addition, a function analysis was performed, whereby the DEPs were classified as involved in photosynthesis, morphogenesis, metabolism, or the stress response. A total of eight proteins were identified as associated with the morphogenesis of stamen development, such as stamen-specific protein FIL1-like (XP_018830780.1), putative leucine-rich repeat receptor-like serine/threonine-protein kinase At2g24130 (XP_018822513.1), cytochrome P450 704B1-like isoform X2 (XP_018845266.1), ervatamin-B-like (XP_018824181.1), probable glucan endo-1,3-beta-glucosidase A6 (XP_018844051.1), pathogenesis-related protein 5-like (XP_018835774.1), GDSL esterase/lipase At5g22810-like (XP_018833146.1), and fatty acyl-CoA reductase 2 (XP_018848853.1). Our results predict several crucial proteins and deepen the understanding of the biochemical mechanism that regulates the formation of male and female flower buds in walnuts.

## 1. Introduction

The Persian walnut (*Juglans regia* L.; also known as the English walnut or common walnut) belongs to the Juglandaceae family, the *Juglans* genus, and order Fagales. Walnut is a monoecious plant; however, the opening time of male and female flowers is not the same, which is defined as dichogamy [1]. From the perspective of plant evolution, the simultaneous opening of male and female flowers can reduce gender interference from their own pollen or stigma, and can increase the probability of external mating to promote heterozygosity [2,3]. Heterodichogamy is a transitional form of evolution towards the dioecious state, which promotes cross-pollination (allogamy) to ensure the diversity of hybrid offspring, thereby enhancing its environmental adaptability and sustainability [4,5]. In addition, the phenomena of male sterility, self-incompatibility, and heterostyly in plants are also aimed at promoting heterozygosity [6,7]. The male and female flowering periods of walnuts can be highly inherited, but the flowering time is influenced by the environment, including the section of *Juglans*, *Carya*, and *Cyclocarya* [8,9,10]. A previous study has suggested that the mating types of heterodichogamy are heritable in *J. regia*, *J. hindsii*, *C. illinoensis*, and *J. mandshurica*, where there were two dominant/recessive alleles at a single locus, with either the protandry being homozygote recessive (gg) or the protogyny being homozygote dominant (GG), or heterozygote (Gg) [11,12].

According to the dichogamy phenomenon of heterodichogamous flowering plants of the same mating type, there are variations in the differentiation and development process of male and female flowers. The genetic mechanism of plant sex determination suggests that hermaphroditic and monoecious plants are the sex determination type of genotype×environment interaction, such as cucumber (*Cucumis sativus*); meanwhile, dioecious plants involve genotype sex determination, such as red bayberry (*Morella rubra*) [13,14]. Genotype sex determination refers to the separation of one or more alleles that determine the gender of a population; however, environmental sex determination (ESD) includes the participation of temperature, photoperiod, light quality, and light intensity in the process of regulation [15,16]. The morphological studies have indicated that the sex determination of unisexual flowers is caused by the selective induction or abortion of sexual organ primordia. The plant sexual system is principally determined by a pair of sex chromosomes, XY or ZW [17]. In the early stages of flower bud differentiation, each flower has the formation of stamens and pistil primordia. However, during the process of organ differentiation, the occurrence of unisexual flowers is caused by the selective abortion of either stamens or pistils in the primordia of bisexual flower flowers [18,19]. The existing research has shown that the role of sex determination genes can occur at various stages of sexual organ development in different plant systems. In the grape (*Vitis vinifera*), three alleles account for the sex phenotypes, including the male gene (*M*), female gene (*F*), and monoecy gene (*H*). The *M* gene is partially dominant to the recessive *H* and *F* allele; then, *FF* plants are gynodioecious, *MF* or *MH* plants are androdioecious, and *HF* or *HH* plants are monoecious [20]. In Caucasian persimmon (*Diospyros lotus*), sex determination is governed by the gene Y-specific sex-determinant candidate, *OGI*, that is specifically displayed in the male flower [21]. The *MeGI* gene is located on the autosome and is only expressed in female flowers; additionally, it can inhibit stamen development. Then, the *OGI* determines the gender of Caucasian persimmon through the regulation of the expression of the *MeGI* gene. *SyGI* and *FrBy* are sex-determining genes both located on the Y-specific region of the genome in kiwifruit (*Actinidia* spp.). *SyGI* suppresses carpel development and *FrBy* promotes male flower development [22,23]. A digital transcriptome expression analysis in the papaya (*Carica papaya*) identifies that the *SVP-like*, as a MADS-box gene, is a candidate gene for sex determination [24]. Based on the analysis of gene expression differences and alternative splicing (AS), the short vegetative phase-like (*CpSVPL*), the chromatin assembly factor 1 subunit A-like (*CpCAF1AL*), and the somatic embryogenesis receptor kinase (*CpSERK*) are the sex determination genes [25].

Floral induction is a critical transition in higher plants, during which the process from vegetative to reproductive growth is regulated by multiple genes [26]. *FLOWERING LOCUS T* (*FT*), *FLOWERING LOCUS C* (*FLC*), *FLOWERING LOCUS F* (*FLF*), *SUPPRESSOR OF OVEREXPRESSION OF CONSTANS 1*(*SOC1*), *LEAFY* (*LFY*), *APETALA1* (*AP1*), *FRUITFUL* (*FUL*), and *SEPALLATA3* (*SEP3*) are transcription factors in *Arabidopsis*, which regulate the floral transition through the transport of signals [27,28,29,30,31,32,33,34]. Genome-wide association studies and transcriptome analyses of walnut have indicated that the *F-box* gene family, as an original gene family, is highly correlated with the maturity of flowers [35]. The transcriptome sequencing results show that a total of 11 regulatory genes involved in the mechanism of hermaphroditism in walnut have been identified, including *FT-like*, *FUL-like*, *SOC1-like*, *SPL-like*, *SVP-like*, *AGL24-like*, *AP1-like*, *LEAFY-like*, *FLC-like*, *FD-like*, and *DELLA-like*. Through the utilization of qRT-PCR, the expression patterns of walnut *JrFUL*, *JrAP1*, *JrLFY*, and *JrSOC1* genes in female and male flower buds of two walnut varieties at different developmental stages led to inferences that the *JrFUL* and *JrAP1* genes inhibit the opening of female walnut flowers, but the *JrLFY* or *JrSOC1* gene function is the opposite [36]. The characterization of *JrLFY* and *JrFT* genes isolated from flower buds from *Juglans regia* L. was achieved, identifying them as homologs of *FLORICAULA* (*LFY*) and *FLOWERING LOCUS T* (*FT*) [37,38].

Proteomics has become a common tool for studying plant physiology and biochemical processes through studying the protein structure and function. During the development of floral organs in Arabidopsis, it is necessary for the plant to consume substances and energy, as well as various hormones such as gibberellins (GAs), auxins (IAAs), cytokinins (CTKs), and abscisic acid (ABA) [39]. The pathways such as the metabolism of hormones, carbon, and energy can be identified related to flower organ development, and stress-related proteins can also be identified [40,41,42]. In *Brassica napus*, stigma-development-related proteins mainly participate in metabolic processes, responses to stimuli or stress, and transport [43]. Using two-dimensional polyacrylamide gel electrophoresis, a large number of stress-related proteins, such as peroxidase, catalase, and heat shock proteins (HSPs), are identified during the petal development of rose (*Rosa hybrida*) [44]. A proteomic identification analysis using two-dimensional electrophoresis in *P. chinensis* illustrated that ascorbate peroxidase (APX), phosphoglycerate kinase (PGK), eukaryotic translation initiation factor 5A2, and temperature-induced lipocalin (TIL) are the proteins related to flowering [45]. During pollen maturation, the development of the surrounding anther somatic cell layers provides nutrients for that process, which leads to significant changes in the morphology of the pollen wall [46,47,48,49]. Male-biased proteins associated with pollen germination and tube growth have been identified in dioecious *C. grandis* [50]. In *Arabidopsis*, approximately 135 differential proteins have been identified from mature pollen. Among them, the glycosyl hydrolase family protein (At2g16730), germin-like protein (GLP8, putative oxalate oxidase, At3g05930), pectin methylesterase inhibitor family protein (At4g24640), actin 4 (At5g59370), and actin-depolymerizing factors (At4g25590, At5g52360) are pollen-specific proteins for cell structure [51]. In the development of the male and female flower buds of *Zanthoxylum planispinum* var. *dintanensis*, a proteomic analysis showed fatty acyl-CoA reductase, pectate_lyase_3 domain-containing protein, and alpha-galactosidase participating in pollen exine synthesis, pollen development, and pollen wall assembly [52]. However, few studies have been performed on the proteomic characteristics during the flora bud maturation processes in walnut. In our study, in order to analyze the sex determination mechanism and reproductive organ development of the proteome, a total of 885 unique differentially expressed proteins (DEPs) were identified by iTRAQ before and after the opening of male and female buds.

## 2. Results

### 2.1. Proteomic Identification of Floral Buds before and after Opening in Walnut

In the four contrasts, 3775 proteins were obtained through iTRAQ with a confidence level of 95%. Among them, 3540 proteins were successfully labeled. In S2/S1, 143 DEPs that consisted of 99 up-regulated proteins (S2/S1 ratio > 1.5, *p* value < 0.01) and 44 down-regulated proteins (S2/S1 ratio < 0.67, *p* value < 0.01) were identified (Figure 1). In S3/S1, 428 DEPs that consisted of 272 up-regulated proteins (S3:S1 ratio > 1.5, *p* value < 0.01) and 156 down-regulated proteins (S3/S1 ratio < 0.67, *p* value < 0.01) were identified. In S4/S2, 479 DEPs that consisted of 284 up-regulated proteins (S4/S2 ratio > 1.5, *p* value < 0.01) and 195 down-regulated proteins (S4/S2 ratio < 0.67, *p* value < 0.01) were identified. In S4/S3, 388 DEPs that consisted of 189 up-regulated proteins (S4/S2 ratio > 1.5, *p* value < 0.01) and 199 down-regulated proteins (S4/S2 ratio < 0.67, *p* value < 0.01) were identified. In these sets, the up-regulated DEPs were more abundant than the down-regulated DEPs. The distribution of DEPs is shown in a Venn diagram (Figure 2). After duplicates were removed, 885 unique DEPs were obtained, and 12 proteins were differentially expressed in all of the examined contrasts (Figure 2, Table 1).

The top 10 up-regulated and down-regulated DEPs are shown on the basis of the fold change in the expression values to survey the DEPs with a low and high abundance (Table 2). Most up-regulated DEPs are associated with the generation of precursor metabolites and energy, photosynthesis, the response to metal ions, the response to stimuli, oxidation reduction, developmental growth, the chloroplast structure, embryogenesis, and stress induction and are inhibitors and stamen-specific. In contrast, down-regulated DEPs are generally associated with DNA replication, RNA methylation, response to high light intensity, response to hydrogen peroxide, protein folding, the metabolism of organic substances, cytokinesis, chloroplast structure, ribosomes, enhancing mRNA, cell death, and dicarboxylic acid biosynthesis. This indicates that the biological functions of up-regulated and down-regulated DEPs are widely distributed.

### 2.2. GO and KEGG Analysis of the Differential Expression Proteins in the Four Contrasts

For an overview of the function of all the detected differential expression proteins (DEPs) and the potential linkage between them, GO and KEGG enrichment analyses were performed for the DEPs detected in the four contrasts. The DEPs were classified into three categories: the biological process (BP), cell component (CC), and molecular function (MF). A total of 1421 GO terms were assigned to 142 DEPs in S2/S1, 1940 GO terms to 403 DEPs in S3/S1, 2064 GO terms to 455 DEPs in S4/S2, and 1935 GO terms to 272 DEPs in S4/S3. The top 10 GO terms in the three GO categories, the “biological process”, “cell component”, and “molecular function”, were counted on the basis of the *p* value in ascending order (Figure 3). Most of the GO terms relating to the “biological process” and “cell component” categories were similar in the four contrasts. For example, the “single-organism metabolic process” and “small-molecule metabolic process” terms in the “biological process” category contained the largest DEP numbers in S3/S1 (235 and 155 DEPs, respectively), S4/S2 (279 and 182 DEPs, respectively), and S4/S3 (215 and 142 DEPs, respectively). For S2/S1, the “small-molecule metabolic process” term (63 DEPs) of the “biological process” contained the largest DEP numbers, followed by the “catabolic process” (54 DEPs) and “carboxylic acid metabolic process” (46 DEPs). For the “cell component” category, the top two largest DEP numbers of S2/S1 and S4/S2 were observed in the “intracellular part” and “cytoplasm”. In S3/S1 and S3/S2, the “cell part” and “cell” contained the largest numbers. In addition, the GO terms of the “molecular function” category in S3/S1, S4/S2, and S4/S3 were “catalytic activity” (275, 322, and 251 DEPs, respectively) and “oxidoreductase activity” (78, 85, and 61 DEPs, respectively), which contained the largest DEP number. For S2/S1, “metal binding” and “oxidoreductase activity” contained the largest DEPs (43 and 27 DEPs, respectively). However, the DEPs were predominantly binding proteins that were involved in molecular function in each contrast.

A total of 142, 403, 455, and 353 DEPs were annotated and assigned to 66 biological pathways in S2/S1, 88 in S3/S1, 94 in S4/S2, and 96 in S4/S2, respectively. Among them, there were 14, 30, 34, and 27 pathways with significant differences (*p* value < 0.05) in each contrast. All of them were placed in ascending order and are illustrated in Figure 4. Six pathways, i.e., “metabolic pathways”, “carbon metabolism”, “biosynthesis of secondary metabolites”, “carbon fixation in photosynthetic organisms”, “pyruvate metabolism”, and “citrate cycle (TCA cycle)”, were simultaneously present with significant differences (*p* value < 0.05) in the four contrasts, as shown in Figure 5. Some paths were unique, i.e., “photosynthesis”, “one carbon pool by folate” for S2/S1, “arginine biosynthesis”, and “tyrosine metabolism” for S3/S1 and “ascorbate and aldarate metabolism” and “beta-Alanine metabolism” for S4/S2. This indicated that the DEPs were involved in pathways that were simultaneously identical and unique.

### 2.3. Protein–Protein Interaction (PPI)

Through a string network analysis in the *Juglans regia* database, the DEPs of XP_018841899.1 (LOC109006913) were discovered to interact with another 10 proteins (Figure 6). The protein of XP_018841899.1 was predicted to be a chlorophyll a-b binding protein of LUCI type 1-like involved in photosynthesis. Among the proteins that interacted with it, five proteins (LOC108991907, LOC109001908, LOC109003241, LOC109001370, LOC108994925) had similar functions, while the remaining three proteins (LOC109016022, LOC109003713, LOC109020369) functioned in the photosystem I reaction center. The protein of XP_018832838.1 (LOC109000426) was up-regulated in S2/S1. From Figure 7, it can be seen that the protein of XP_018830780.1 is the unique protein in S3/S1 that is related to stamen-specific expression. The four proteins (LOC109015525, LOC109019583, LOC109006417, LOC109012439) that interact with it are involved in nitrogen metabolism and energy metabolism. One protein (LOC108984596) has the function of an aquaporin.

## 3. Discussion

### 3.1. Changes in Floral Bud Characteristics before and after Opening in Walnut

Walnut flower bud differentiation is a discontinuous process with a cycle of about one year. There is a stable correlation between the external morphological characteristics of male and female flower buds and internal structural changes [53,54]. The periods of flower bud opening before and after in walnut are male flower and female formation stages. During this period, the external morphology of the flower buds gradually expands. The outer scales of the female flower buds are lignified, and the male flower bud inflorescences are gradually loosened (Yu et al., 2022) [55]. At the time, the pollen mother cells are differentiating and forming, and this is also an important period for the differentiation of the walnut pistils and anthers. During this period, the germination and expansion of flower buds and the pumping of new shoots and leaf development all require a significant amount of nutrients. There were many factors affecting the differentiation process of flower buds, among which the levels of temperature and light played a significant role [56,57].

### 3.2. Common DEPs Play Promotional Roles in All Four Contrasts

Among all DEPs, there were 12 proteins detected in all four contrasts (Table 1). A GO and KEGG analysis of the 12 common DEPs showed that their functions are distributed in the cytoplasm metabolic pathways, photosynthesis, glyoxylate and dicarboxylate metabolism, and the biosynthesis of secondary metabolites, which are involved in energy production and conversion, synthesis, and the breakdown of proteomes.

Formate dehydrogenase (FDH) is a widely distributed enzyme that relies on NAD^+^ in organisms, which catalyzes the oxidation of formic acid to carbon dioxide and reduces NAD^+^ to NADH in the process. Formic acid is involved in one-carbon metabolism, which is essential for all living organisms, providing the C^1^ units necessary for nucleotide synthesis, mitochondrial and chloroplast protein synthesis, amino acid metabolism, methyl synthesis, and vitamin metabolism [58]. The xyloglucan endotransglucosylase/hydrolase protein displays activities in relation to xyloglucan endotransglucosylase (XET) and xyloglucan endohydrolase (XEH). The previous studies in this area have suggested that the XEH gene plays an important role in flower formation. The XEH gene is involved in the formation of pistil buds in rapeseed (*Brassica napus*) [59], the elongation of stamens in Arabidopsis (*Arabidopsis thaliana*) [60], and the morphogenesis of petals in boat orchids (*Cymbidium* ssp.) [61]. Sedoheptulose-1,7-bisphosphatase (SBPase) is a Calvin cycle enzyme and functions in the fixation of photosynthetic carbon. In the plants of transgenic tobacco (*Nicotiana tabacum* L. cv. Samsun) [62], tomato (*Solanum lycopersicum*) [63], Arabidopsis (*Arabidopsis thaliana*) [64], and upland cotton (*Gossypium hirsutum*) [65], an increase in SBPase activity can promote an increase in the photosynthetic rate, as well as a significant increase in sucrose and starch accumulation. The main function of light harvesting chlorophyll a/b binding proteins (LHCs) is to absorb light energy and then transfer it to the reaction center. In tomato, the changes in the amount of the chlorophyll a/b binding protein complex (LHC II) captured by light are crucial for regulating the absorption of light by photosystem II [66].

### 3.3. Protein Species Expressed Specifically Related to Photosynthesis in Floral Bud Development

Some differentially expressed proteins were identified in the development stage of flower buds in the four contrasts, such as pentatricopeptide repeat-containing protein At1g02060, chloroplastic-like (XP_018852740.1), oxygen-evolving enhancer protein 2 (XP_018832838.1), ribulose bisphosphate carboxylase/oxygenase activase, chloroplastic-like isoform X1 (XP_018814447.1), polyphenol oxidase, chloroplastic-like (XP_018858848.1), photosystem II 47 kDa protein (YP_009186194.1), and early nodulin-like protein 1 (XP_018833667.1).

Chloroplasts are essential photosynthetic organelles in plants. In higher plants, many pentapeptide repeat (PPR) proteins located in chloroplasts have become the main nuclear factors involved in chloroplast gene expression and RNA metabolism. In *Arabidopsis*, a sequence analysis has shown that the chloroplast protein encoded by *pigment-defective mutant 3* (*PDM3*) consists of 12 pentatricopeptide repeat domains and plays an important role in chloroplast development [67]. In *C. reinhardtii,* oxygen evolution enhancer protein 2 (OEE2) plays a crucial role in the assembly/stability of photosystem II (PS II) complexes as an important protein for photosynthetic oxygen evolution [68]. OEE2 of PS II, as a chloroplast protein, is predicted to be involved in the pathways of the “Calvin cycle” and “photosynthesis” [69]. In Arabidopsis, OEE2 activity is additionally regulated by AtGRP-3/WAK1, as a molecule downstream of AtGRP-3/WAK1, and possibly in defense signaling [70].

Ribulose bisphosphate carboxylase/oxygenase (RuBisCO) activase (RCA) is a soluble chloroplast protein encoded by nuclear genes. RCA is located at the intersection of two opposite but interrelated cycles of either photosynthetic carbon reduction or carbon oxidation, and has been identified as the most important factor determining the net photosynthetic rate [71]. From a functional perspective, RCA shares similarities with molecular chaperones. Based on sequence similarity, RCA belongs to the ATPase related to various cellular activities’ (AAA+) protein family. Ribulose-1,5-bis-phosphate carboxylase/oxygenase (Rubisco) acts as a rate-limiting enzyme for carbon fixation during photosynthesis. RUBISCO ACCUMULATION FACTOR 1 (RAF1) is a chloroplast protein that is mainly expressed in the bundle sheath chloroplasts, and is essential for the accumulation of RUBISCO in maize (*Zea mays*), showing consistent functions [72].

Polyphenol oxidases (PPOs) are nuclear-encoded chloroplast proteins located in the chloroplast thylakoid and other types of a plastid matrix. PPOs may play an energy conversion role in chloroplasts, transmitting the molecular oxygen generated via photoreduction [73]. The 47 kDa antenna chlorophyll protein has previously been isolated from spinach (*Spinacia oleracea*) photosystem II [74].

In the early stages of nitrogen-fixing root nodule development in leguminous plants, early nodulation protein (ENOD)-like proteins are expressed and belong to the subclass of plant anthocyanins. *BcBCP1* is a gene encoding the early nodulation protein (ENOD)-like proteins in plant anthocyanins, and its expression is both induced and inhibited via external stimuli, such as salinity, abscisic acid, low temperatures, and methyljasmonic acid [75].

### 3.4. Protein Species Expressed Specifically Related to Morphogenesis in Floral Bud Development

Some differentially expressed proteins were identified in the flower bud development stage in the four contrasts, such as stamen-specific protein FIL1-like (XP_018830780.1), putative leucine-rich repeat receptor-like serine/threonine-protein kinase At2g24130 (XP_018822513.1), cytochrome P450 704B1-like isoform X2 (XP_018845266.1), ervatamin-B-like (XP_018824181.1), probable glucan endo-1,3-beta-glucosidase A6 (XP_018844051.1), pathogenesis-related protein 5-like (XP_018835774.1), GDSL esterase/lipase At5g22810-like (XP_018833146.1), and fatty acyl-CoA reductase 2 (XP_018848853.1).

Stamens, as the male reproductive organ, produce and disperse pollen to ensure the successful breeding of offspring. The development of stamens is a complex and orderly process, which in flowering plants is regulated by different transcription factors at different stages. Four stamen-specific proteins (SSPs) have been identified from buffel grass (*Cenchrus ciliaris* L.) [76]. Two of these proteins are specifically expressed during the microsporogenesis and mononuclear microsporogenesis stage. The results indicated that SSPs may be involved in the sexual differentiation of this species, and may also be a specific protein for gametophyte or meiosis.

Leucine-rich repeat (LRR) receptor kinases (LRR-RKs) are the largest family of receptor kinases in plants. The ERECTA (ER) subfamily of leucine-rich repeat (LRR) receptor kinases (LRR-RKs) plays a crucial role in plant development, mainly in regulating cell division, organ morphology, and inflorescence composition in Arabidopsis. The *OsERL* of Arabidopsis and rice is expressed in sporophytic and tapetal cells in the anther and plays a critical role in regulating the events that determine the periclinal divisions in anthers [77,78]. Plant cytochrome P450 monooxygenases (CYPs) belong to a huge superfamily of heme-protease enzymes and participate in many primary and secondary metabolism reactions. *CYP704B1* was identified as being expressed in the developing anthers that regulate the synthesis of pollen sporopollenin in *Arabidopsis* [79]. *BoCYP704B1* is located in the endoplasmic reticulum and was identified as being involved in regulating male infertility. In addition, *BoCYP704B1* is highly expressed at the tetrad and in the early stages of monocytes in cabbage anther tapetum development [80]. GDSL lipase is involved in regulating the formation of maize anthers, and its coding gene (*ZmMs30*) is specifically expressed [81]. Ervatamin B, purified from the latex of *Ervatamia coronaria*, is a cysteine protease (CP) of the papain superfamily [82]. Cysteine proteases are an important class of proteolytic enzymes involved in the programmed cell death (PCD) process of the anther tapetum during the development of many plant organs [83,84].

Glucan endo-1,3-beta-glucosidases are involved in the metabolism of female flower cell walls and the development of male flowers [85]. Pathogenesis-related proteins (PRs) are mainly distributed in plant vacuoles and intercellular spaces, produced by plants in pathological or pathological-related environments, encoded by multiple genes, and widely present in higher plants [86]. The accumulation of PRs could be induced via pathogenic factors (viruses, bacteria, fungi, etc.), physiological changes (nutrient deficiency, flower bud differentiation, callus formation, cell death, etc.), chemical signals (plant hormones, amino acid derivatives, etc.), and physical signals (rays, external damage, high temperatures, etc.). A PR-2-type protein has been proven to significantly promote pollen development in tobacco [87]. *GELP77* (AT4G10950) is a GDSL-type esterase/lipase gene involved in pollen development, especially in pollen dissociation and microspore nucleus development, and its absence can lead to male infertility in *Arabidopsis* [88]. The GDSL esterase/lipase protein (GELP) is also required for anther and pollen development in rice [89]. In flowering plants, fatty alcohols and their derivatives are necessary for the formation of the anther stratum corneum and pollen wall during pollen development [90]. Fatty acyl-CoA reductases (FARs) are key enzymes in the catalytic process in *Arabidopsis* that forms primary fatty alcohols from fatty acyl-CoAs or fatty acid-acyl carrier proteins [91]. Based on similar structural domains, it has been confirmed that the FAR gene encodes MALE STILITY2 in *Arabidopsis*, and its protein homologue defective pollen wall (DPW) also performs similar functions in rice [92,93].

### 3.5. Protein Species Expressed Specifically Related to Metabolism in Floral Bud Development

Some of the identified differentially expressed proteins participated in metabolism in the flower bud development stage in the four contrasts, such as UDP-glycosyltransferase 79B30 isoform X1 (XP_018838485.1), 4-coumarate-CoA ligase-like 1 (XP_018836778.1), exopolygalacturonase-like (XP_018849923.1), mannan endo-1,4-beta-mannosidase 4-like (XP_018843119.1), DEAD-box ATP-dependent RNA helicase 3, chloroplastic-like (XP_018807936.1), aspartyl protease AED3-like (XP_018818184.1), 60S ribosomal protein L11 (XP_018857107.1), bifunctional 3-dehydroquinate dehydratase/shikimate dehydrogenase, and chloroplastic-like (XP_018847629.1).

4-Coumarate: The coenzyme A ligase (4CL) is one of the key enzymes in the phenylpropanoid metabolism pathway. 4CL is involved in the regulation of many metabolites, including lignin, flavonoids, lignans, phenylpropionate-like, hydroxycinnamic acid amides, and sporopollenin. 4CL is essential for the metabolism of various substances during plant growth, and this gene family is involved in various metabolic pathways including lipid metabolism and amino acid metabolism in *Bletilla striata* [94]. In addition, the *os4CL2* gene is essential for rice (*Oryza japonica*) anthers and may also play a role in the formation of flavonoids [95]. The results of the transcriptomic analysis using RNA sequencing (RNA-seq) indicate that the *Pp4CL* of peach (*Prunus persica*) regulates pollen fertility through the phenylalanine metabolic pathway [96]. Glycosyltransferases (GTs) are widely involved in various small-molecule sugar metabolism processes in plants [97]. The UDP glycosyltransferases (UGTs), often referred to as 1 glycosyltransferases (GT1s), of at least 92 families are the largest superfamily in plants. Thus, plant UGTs are classified within 30 families between UGT71 and UGT100 [98].

Polygalacturonase (PG) plays an important role in the degradation of plant cell walls through catalyzing the breakdown of pectin, according to its different degradation pathways, which can be divided into endo-polygalacturonase (endoPG), exopolygalacturonase (exoPG), and rhamnopolygalacturonase (rhamnoPG) [99]. Previous, extensive research showed that PG plays an important role in fruit development, ripening, and softening. The exopolygalacturonase (exoPG) activities of cell-wall-modifying enzymes are increased in the melting-flesh peach during ripening, induced by ethylene [100]. Most of the identified *ZjPG* members in Chinese jujube (*Ziziphus zizyphus* Mill) have a relatively high expression in flowers and fruits, which may be involved in the substantial formation of jujube fruits [101]. In addition, PG genes play an important role in a series of physiological and biochemical processes in plants, such as the response to external environmental stimuli, anther dehiscence, pollen grain maturation, and organ shedding [102,103,104]. Mannan endo-1,4-beta-mannosidase (MAN) is significantly differentially expressed in relation to fruit cracking in tomato [105] and was predicted to be involved in the metabolism of cell wall degrading enzymes.

DEAD-box RNA helicases are special RNA molecular chaperones that participate in RNA metabolism, including precursor RNA splicing, ribosome synthesis, RNA degradation, and gene expression. The *HEN2* gene of Arabidopsis (*Arabidopsis thaliana*) encoding a DExH boxRNA helicase is specifically expressed in the floral meristem and developing flowers, and plays a role in flower development [106]. Zhang et al. [107] identified that the DEAD-box RNA helicase was up-regulated in the late pollen development stage of maize (*Zea mays*).

Aspartyl protease encoded by *AED3* (APOPLASTIC, *EDS1*-DEPENDENT) was identified as participating in the protein metabolism in Arabidopsis [108]. Ribosomal proteins are encoded by small gene families in plants [109]. Through reliance on the combination of chaperonin protein 60 (cpn60) and 10 kDa chaperonin protein (cpn10), molecular chaperones assist the folding of synthesized proteins and maintain their stability [110]. Research has suggested that the *RPL11* gene is involved in the development of pollen and anthers in *Arabidopsis* [111,112]. The bifunctional 3-dehydroquinate dehydratase/shikimate dehydrogenase (DHD/SHD) is an important enzyme in the shikimate pathway, involved in the synthesis of secondary metabolites in plants [113].

### 3.6. Protein Species Expressed Specifically Related to Stress Response in Floral Buds’ Development

Some of the identified differentially expressed proteins participate in the stress response in the flower bud development stage in the four obtained contrasts, such as late embryogenesis abundant protein 2-like (XP_018834136.1) and osmotin-like protein isoform X1 (XP_018823302.1).

Late embryogenesis abundant proteins 2 (LEA 2 proteins) are usually present in plant dormant seeds or accumulate in significant amounts in plant tissues when subjected to external stress [114,115]. LEA 2 proteins play a role as molecular chaperones and hydrophilic solutes in metabolism, preventing protein aggregation, stabilizing, and protecting the protein structure and function under water stress [116,117]. The accumulation of LEA 2 proteins during the opening process of female and male walnut flowers is estimated to be related to seed embryo development and the external environment.

Osmotin and osmotin-like proteins (OLPs) are produced in plants when they experience biotic or abiotic stress. Research has found that OLPs are highly expressed in the flowers and fruit of tomato (*Lycopersicon esculentum*) and accumulate in significant amounts in the pistils during flowering [118]. Late embryogenesis abundant (LEA) and osmotin and osmotin-like proteins (OLPs) are products of the cellular stress response, and are produced through a series of complex cellular physiological and biochemical changes and play a role in maintaining cellular balance.

## 4. Materials and Methods

### 4.1. Plant Materials

The plant materials were collected from the National Horticultural Germplasm Resources Center’s walnut variety nursery in Zhengzhou, China (E113°22′14.8″, N34°579.78″) from February to March. The walnut variety named ‘Zhuxing’ was protogynous, 8 years old, and grew well. The samples were used in the period before and after the opening of the female and male flower buds, which were collected in four states, namely, unopened pistils (the stigma is closed and the pollen cannot be accepted), opened pistils (the stigma is open and pollen can be accepted), unopened stamens (the spikes are less than 30 mm long and unpowdered), and opened stamens (the spikes are longer than 100 mm long and are already powdered), labeled S1, S2, S3, and S4, respectively (Figure 8). All inflorescence samples were immediately frozen in liquid nitrogen and stored at −80 °C until use. Ten biological replicates were mixed and then divided into the three prepared experimental replicates.

### 4.2. Protein Extraction and Quantization

A total of 1 g of a walnut sample was weighed and ground in liquid nitrogen. TCA/pre-cooled acetone (containing 0.1% DTT and 1 mM PMSF) was added and left to precipitate at −20 °C for more than 3 h. Centrifugation at a speed of 15,000 rpm/min at 4 °C was performed for 20 min; the supernatant was removed and the sediments were retained. Then, the previous step was repeated until the sediments turned white. A cracking solution was added to the sediments and ultrasonic treatment was performed for 5 min to aid dissolution. After that, centrifugation at 20,000 rpm/min was performed for 30 min to obtain the supernatants. Four times the volume of the sample solution with pre-cooled acetone was added and left to precipitate at −20 °C for more than 2 h. The sediments were dried in a vacuum freeze dryer for 30 min and then stored in a −80 °C refrigerator. The protein concentration was measured using a Bradford Assay Kit and the amount of protein required for the iTRAQ analysis was determined.

### 4.3. Protein Hydrolysis and iTRAQ Labeling

The protein hydrolysis procedure followed the operating plan used by Wisniewski et al. [119]. For each sample, 100 μg of protein was added to five volumes of pre-cooled acetone, and then the mixture was stewed at −20 °C for more than 1 h. Next, the samples were centrifuged at 15,000 rpm for 20 min at 4 °C, and the deposits were collected and dried using a vacuum freeze dryer. Then, 50 μL of a dissolution buffer was added from the iTRAQ Kit and fully vortexed; after mixing well, 4 μL of a reducing reagent was added and the solution was allowed to stand at 60 °C for 1 h. Next, 2 μL of cysteine-blocking reagents was added to each sample and they were allowed to stand at room temperature for 10 min. The protein solution was centrifuged at 12,000 rpm for 40 min at 4 °C in a 10 KDa Nanosep MF concentration tube. Then, the solution was centrifuged at 12,000 rpm for 30 min at 4 °C three times with 100 µL of a dissolution buffer. Finally, the filter membrane was placed into a new tube, trypsin was added (concentration: 1 μg/μL) to each sample at a 1:40 ratio of enzyme and protein, and the mixture was hydrolyzed at 37 °C for 14 h. The peptide segment was dissolved in 150 µL of ethanol and then labeled using the iTRAQ Reagent 8-plex kit (AB Sciex Inc., Framingham, MA, USA) (please refer to the manufacturer’s instructions for specific operating conditions).

### 4.4. HPLC Separation of Peptide Mixture

After freeze-drying, the samples were fully dissolved in the 100 µL Buffer A (25 mM NaH_2_PO_4_ in 25% acetonitrile, pH 2.7) solution. The peptides were separated in strong cation exchange (SCX) using Agilent 1200 HPLC and detected at wavelengths of 215 nm and 280 nm. The main parameters of a chromatographic column were Poly-SEA 5 µm, 300 Å 2.0 × 150 mm (Michrom Bioresources Inc., Drive Auburn, CA, USA). The samples were collected in a gradient at a flow rate of 0.3 mL/min. One tube was collected within the first 5 min, the next ten tubes were collected every 4 min between 6 and 44 min, and one tube was collected in the last 45–50 min, forming a total of twelve tubes of the sample solution. Then, all solutions were thoroughly freeze-dried.

### 4.5. LC-MS/MS Analysis

The specific method was as follows: The freeze-dried peptide samples were dissolved again using Nano-RPLC Buffer A. Then, the samples were rinsed and desalinated on a C_18_ pretreatment column (100 µm × 3 cm, C_18_, 3 µm, 150 Å) at 2 μL/min for 10 min using Eksigent nanoLC Ultra™ 2D (AB Sciex Inc., MA, USA). The samples were eluted by increasing mobile phase B from 5% to 35% within 70 min using analytical ChromXP column C_18_ (75 μm × 15 cm, C_18_, 3 μm, 120 Å). Mass spectrometry analysis data were obtained using the Triple TOF5600 system (AB Sciex Inc., MA, USA) combined with a Nano spray III source (AB Sciex Inc., MA, USA). The specific parameters were as follows: The spray voltage was 2.5 kV, with pressures of the curtain gas and nebulizer gas of 30PSI and 5PSI, respectively, and the heater temperature was 150 °C. The scanning mode of the mass spectrum was information-dependent acquisition (IDA). The single cycle time of IDA was set to 2.5 s, and the single-map scanning time of single-stage TOF-MS was 250 ms. The secondary IDA charge was set to 2^+^ to 5^+^, with a maximum collection count of 35, and more than 150 per second were counted.

### 4.6. Data Analysis and Reliable Protein Screening

Protein Pilot Software v. 5.0 (AB Sciex Inc., MA, USA) was used to obtain the data via a search of the NCBI *Juglans regia* database (https://www.ncbi.nlm.nih.gov/, accessed on 21 October 2021). When each protein peptide demonstrated a false discovery rate (FDR) of <1% and an unused ProtScore of >1.3 and contained at least one unique peptide with a confidence level above 95%, it was defined as a trusted protein. After performing *t*-tests, the data were considered reliable when the error factor was <2, and when the ratio fold changes were >1.5 (up-regulated) or <0.67 (down-regulated) in both technical replicates; they were selected as the cutoff values for identifying proteins with significant differences.

### 4.7. Bioinformatics Analysis

Through searching the UniProt *Juglans regia* database, the functions of all differentially expressed proteins were thoroughly analyzed (http://www.uniprot.org/, accessed on 31 May 2023). To clarify the biological functional characteristics of differentially expressed proteins, Gene Ontology (GO) (https://www.ebi.ac.uk/QuickGO/, accessed on 31 May 2022) and KEGG (http://www.genome.jp/kegg/pathway.html, accessed on 31 May 2022) were used to perform differential protein analyses on the database. A Venn diagram was mapped using the online data processing tool (http://bioinformatics.psb.ugent.be/webtools/Venn/, accessed on 21 July 2023). The interaction network analysis of proteins was determined using online STRING software 12.0 (https://string-db.org/).

## 5. Conclusions

In this study, 3540 proteins were detected in walnut floral buds before and after opening. On the basis of the RPLC-MS/MS analysis, 885 unique differentially expressed proteins (DEPs) were identified. Among all DEPs, 12 common proteins were detected in all four contrasts involved in the energy production and conversion, photosynthesis, synthesis, and breakdown of the proteomes, such as formate dehydrogenase, probable xyloglucan endotransglucosylase/hydrolase protein 6, sedoheptulose-1,7-bisphosphatase, and chlorophyll a-b binding protein of LHC II type 1. Formate dehydrogenase (FDH), as a widely distributed enzyme located in mitochondria, had an effect on both energy production and conversion. The xyloglucan endotransglucosylase/hydrolase protein (XEH) potentially played a significant role in flower formation, especially in the formation of pistils, stamen elongation, and petal morphogenesis. The functional performance of sedoheptulose 1,7-bisphosphatase (SBPase) depended on chloroplasts, promoting the photosynthetic capacity and carbohydrate accumulation in the Calvin cycle through catalyzing the synthesis of sedoheptulose 7-phosphate. Light harvesting chlorophyll a/b binding (LHCB) proteins are an important carrier for the function of the photosystem II complex, which has an essential role in light capture and photoprotection. Although LHC protein synthesis is controlled by nuclear genes, their function is exerted in the cytoplasmic reticulum. In this study, differential proteins were significantly enriched in the cytoplasm, indicating that they are important sites for proteins related to the synthesis and breakdown of the proteomes. In addition, six DEPs, eight DEPs, eight DEPs, and two DEPs were discovered to be involved in the processes of photosynthesis, morphogenesis, metabolism, and stress response, respectively.

## Figures and Tables

**Figure 1 ijms-25-07878-f001:**
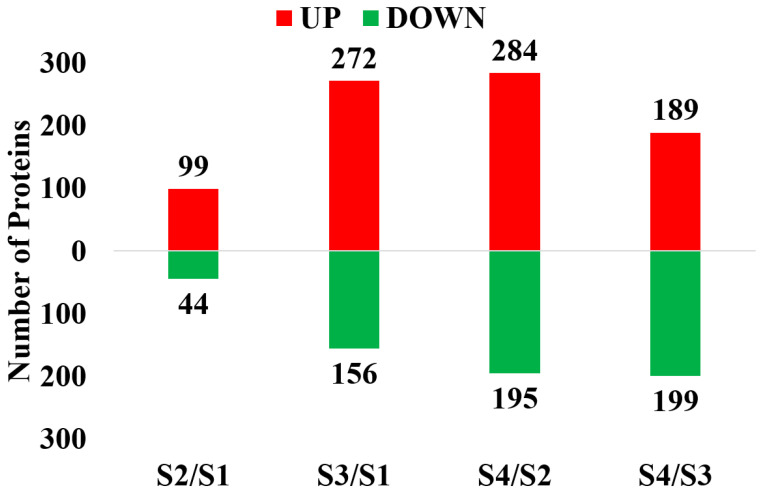
The number of up-regulated and down-regulated DEPs identified in each contrast.

**Figure 2 ijms-25-07878-f002:**
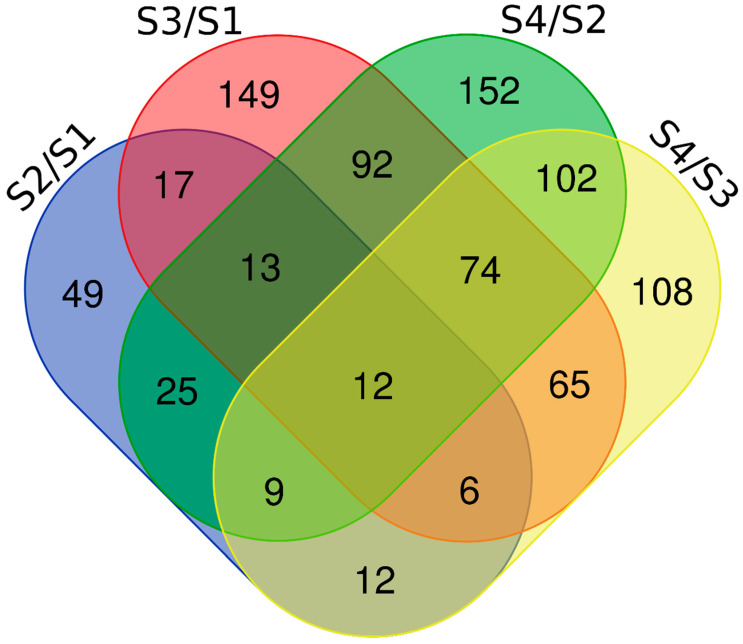
The distribution of DEPs in S2/S1, S3/S2, S4/S2, and S4/S3. A Venn diagram illustrating the specific and common proteins between the different contrasts.

**Figure 3 ijms-25-07878-f003:**
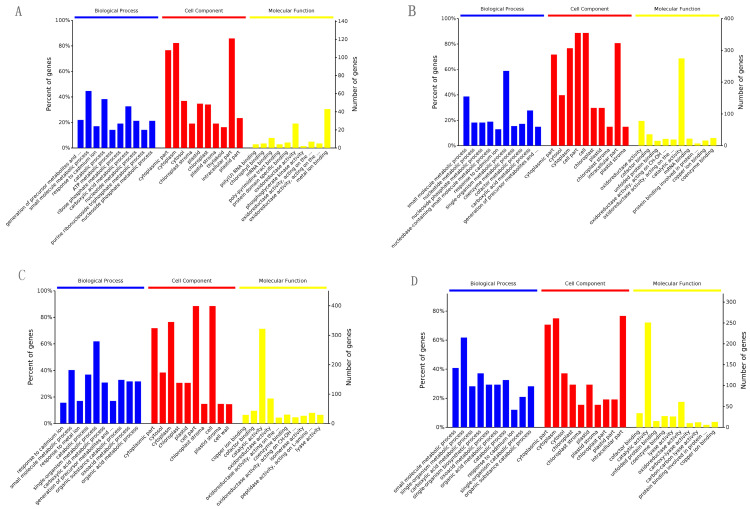
The most significant enriched GO terms of the differentially expressed proteins at each contrast. (**A**), S2/S1; (**B**), S3/S1; (**C**), S4/S2; (**D**), S4/S3. Note: The full name with ellipsis in (**A**) are generation of precursor metabolites and energy, oxidoreductase activity, acting on the CH-OH group of donors, NAD or NADP as acceptor and oxidoreductase activity, acting on the CH-CH group of donors respectively. The full name with ellipsis in (**B**) are generation of precursor metabolites and energy, oxidoreductase activity, acting on the CH-CH group of donors, oxidoreductase activity, acting on the CH-OH group of donors, NAD or NADP as acceptor and protein binding involved in protein folding respectively. The full name with ellipsis in (**C**) are generation of precursor metabolites and energy, oxidoreductase activity, acting on the CH-OH group of donors, NAD or NADP as acceptor, oxidoreductase activity, acting on the CH-CH group of donors and peptidase activity, acting on L-amino acid peptides respectively. The full name with ellipsis in (**D**) is protein binding involved in protein folding.

**Figure 4 ijms-25-07878-f004:**
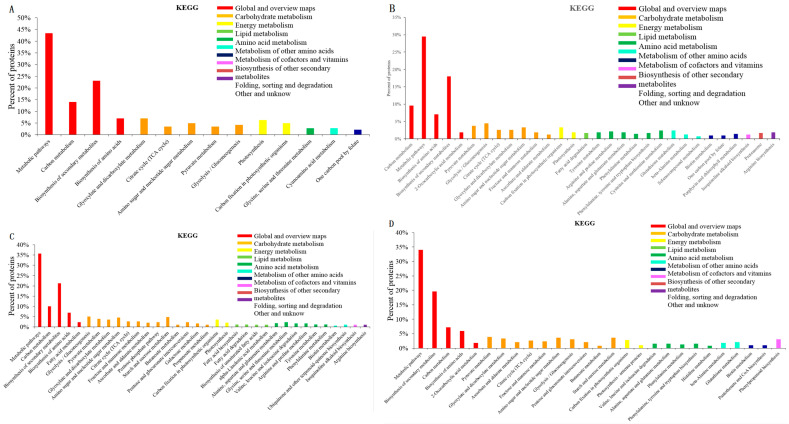
The most significant associated KEGG pathways of differentially expressed proteins at each contrast. (**A**), S2/S1; (**B**), S3/S1; (**C**), S4/S2; (**D**), S4/S3.

**Figure 5 ijms-25-07878-f005:**
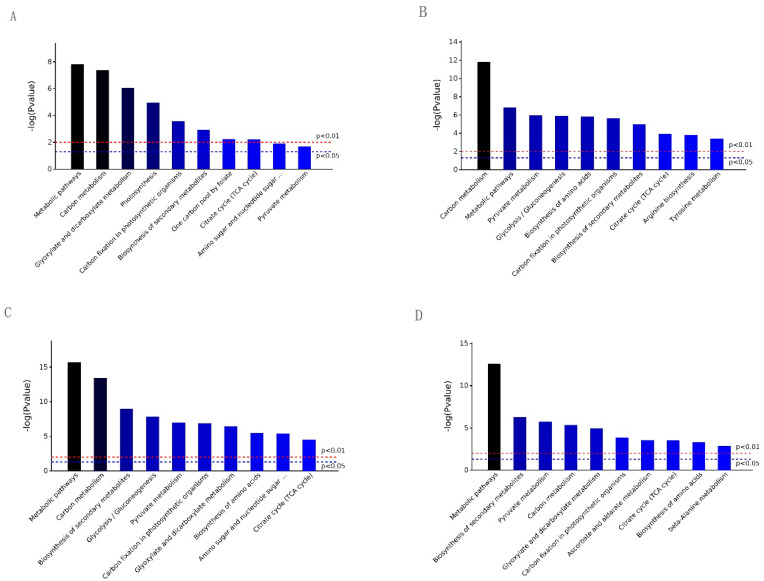
The KEGG classification of the identified DEPs. *p* values below the 0.05 threshold. (**A**), S2/S1; (**B**), S3/S1; (**C**), S4/S2; (**D**), S4/S3. Note: The full names with ellipses in (**A**,**C**) are both Amino sugar and nucleotide sugar metabolism.

**Figure 6 ijms-25-07878-f006:**
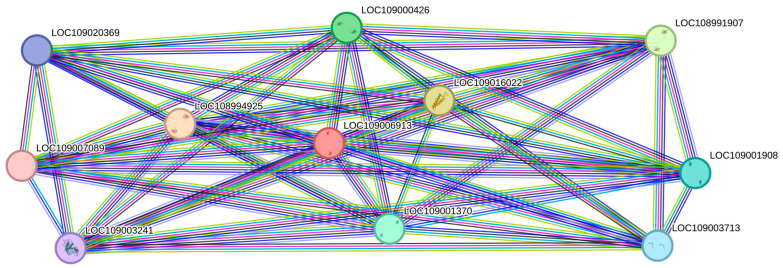
The string networks of the DEPs of XP_018841899.1.

**Figure 7 ijms-25-07878-f007:**
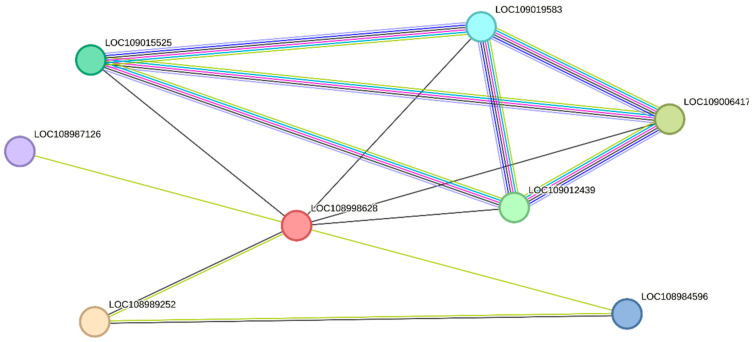
The string networks of the DEPs of XP_018830780.1.

**Figure 8 ijms-25-07878-f008:**
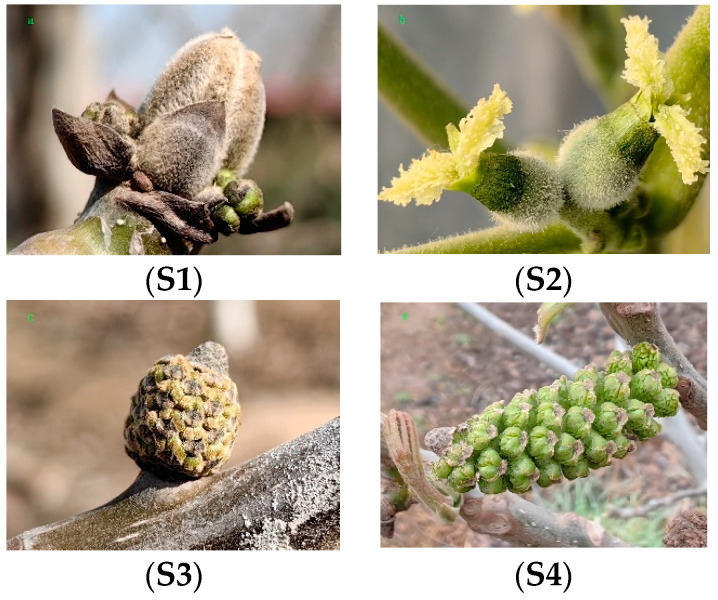
Photographs of the pistils and stamens of the walnut plant cultivar ‘Zhuxing’. Unopened pistils (**S1**); opened pistils (**S2**); unopened stamens (**S3**); opened stamens (**S4**).

**Table 1 ijms-25-07878-t001:** A summary of the common differentially expressed proteins in four contrasts.

Biological Function	Accession Number	Protein Description	*p* Value	Fold Change			
				S2/S1	S3/S1	S4/S2	S4/S3
Single-organism process/post-embryonic development	XP_018834821.1	Stress-induced protein KIN2-like [*Juglans regia*]	4.78 × 10^−13^/2.38 × 10^−3^	2.0669	23.8047	47.2468	4.8394
Carbohydrate metabolic process	XP_018860371.1	Probable xyloglucan endotransglucosylase/hydrolase protein 6 [*Juglans regia*]	8.43 × 10^−21^	2.899	5.8383	7.8709	3.1480
Generation of precursor metabolites and energy/photosynthesis	XP_018841899.1	Chlorophyll a-b binding protein of LHC II type 1-like [*Juglans regia*]	7.25 × 10^−20^/1.17 × 10^−8^	2.3899	2.2833	2.0106	2.1159
Response to cadmium ions	XP_018823000.1	Formate dehydrogenase, mitochondrial [*Juglans regia*]	2.59 × 10^−21^	2.2702	3.0172	2.7081	2.1303
Small-molecule metabolic process/photosynthesis/post-embryonic development	XP_018821822.1	Sedoheptulose-1,7-bisphosphatase, chloroplastic [*Juglans regia*]	1.38 × 10^−36^/1.17 × 10^−8^/2.38 × 10^−3^	2.0007	3.2978	2.8434	1.6835
Catabolic process	XP_018852509.1	Puromycin-sensitive aminopeptidase isoform X1 [*Juglans regia*]	3.90 × 10^−25^	1.5417	1.8803	2.3313	1.9589
Metabolic process	XP_018811660.1	Putative 4-hydroxy-4-methyl-2-oxoglutarate aldolase 2 [*Juglans regia*]	2.56 × 10^−10^	2.9671	2.2085	0.3743	0.4143
Small-molecule metabolic process	XP_018837921.1	1,4-Dihydroxy-2-naphthoyl-CoA synthase, peroxisomal-like isoform X1 [*Juglans regia*]	1.38 × 10^−36^	0.3629	6.7074	2.8236	0.1578
Single-organism cellular process	XP_018809241.1	COBRA-like protein 7 [*Juglans regia*]	7.94 × 10^−14^	0.4089	2.2769	13.4576	3.7633
Small-molecule metabolic process/post-embryonic development	XP_018845399.1	Inositol-3-phosphate synthase-like [*Juglans regia*]	1.38 × 10^−36^/2.38 × 10^−3^	0.4059	2.9501	3.6717	0.5321
―	XP_018845416.1	Uncharacterized protein LOC109009413 [*Juglans regia*]	―	0.4834	2.9261	1.7107	0.3114
Single-organism cellular process	XP_018831299.1	Tubulin-folding cofactor A-like [*Juglans regia*]	7.94 × 10^−14^	0.4295	0.3698	2.2558	2.4488

**Table 2 ijms-25-07878-t002:** List of the top 10 up-regulated and down-regulated DEPs in four contrasts.

Contrast	Accession Number	Protein Description	Fold Change
DEPs (S2/S1)	XP_018832838.1	oxygen-evolving enhancer protein 2, chloroplastic [*Juglans regia*]	4.9200
	XP_018846385.1	probable nucleoredoxin 1 [*Juglans regia*]	4.2982
	XP_018814447.1	ribulose bisphosphate carboxylase/oxygenase activase, chloroplastic-like isoform X1 [*Juglans regia*]	4.0772
	XP_018835529.1	endochitinase-like [*Juglans regia*]	4.0325
	XP_018843857.1	beta-glucosidase 12-like [*Juglans regia*]	4.0188
	XP_018858848.1	polyphenol oxidase, chloroplastic-like [*Juglans regia*]	3.7727
	YP_009186194.1	photosystem II 47 kDa protein (chloroplast) [*Juglans regia*]	3.6141
	XP_018847910.1	phosphoglycerate kinase, chloroplastic [*Juglans regia*]	3.3870
	XP_018834056.1	probable alpha-mannosidase At5g13980 isoform X4 [*Juglans regia*]	3.3830
	XP_018843283.1	profilin [*Juglans regia*]	3.3010
	XP_018831299.1	tubulin-folding cofactor A-like [*Juglans regia*]	0.4295
	XP_018807936.1	DEAD-box ATP-dependent RNA helicase 3, chloroplastic-like [*Juglans regia*]	0.4122
	XP_018809241.1	COBRA-like protein 7 [*Juglans regia*]	0.4089
	XP_018845399.1	inositol-3-phosphate synthase-like [*Juglans regia*]	0.4059
	XP_018842878.1	10 kDa chaperonin-like [*Juglans regia*]	0.4022
	XP_018850608.1	caffeic acid 3-O-methyltransferase [*Juglans regia*]	0.3738
	XP_018828035.1	enhancer of mRNA-decapping protein 4-like [*Juglans regia*]	0.3632
	XP_018837921.1	1,4-dihydroxy-2-naphthoyl-CoA synthase, peroxisomal-like isoform X1 [*Juglans regia*]	0.3630
	XP_018845332.1	aldehyde oxidase GLOX [*Juglans regia*]	0.3007
	XP_018849734.1	UBP1-associated protein 2C-like isoform X3 [*Juglans regia*]	0.2916
DEPs (S3/S1)	XP_018836778.1	4-coumarate-CoA ligase-like 1 [*Juglans regia*]	43.2116
	XP_018852740.1	pentatricopeptide repeat-containing protein At1g02060, chloroplastic-like [*Juglans regia*]	39.9188
	XP_018830780.1	stamen-specific protein FIL1-like [*Juglans regia*]	34.8341
	XP_018848917.1	probable LRR receptor-like serine/threonine-protein kinase At4g36180 [*Juglans regia*]	34.7458
	XP_018822513.1	putative leucine-rich repeat receptor-like serine/threonine-protein kinase At2g24130 [*Juglans regia*]	32.0020
	XP_018848853.1	fatty acyl-CoA reductase 2 [*Juglans regia*]	29.4214
	XP_018844051.1	probable glucan endo-1,3-beta-glucosidase A6 [*Juglans regia*]	28.6322
	XP_018835774.1	pathogenesis-related protein 5-like [*Juglans regia*]	27.6657
	XP_018838485.1	UDP-glycosyltransferase 79B30 isoform X1 [*Juglans regia*]	26.5040
	XP_018833146.1	GDSL esterase/lipase At5g22810-like [*Juglans regia*]	26.3574
	XP_018818895.1	interactor of constitutive active ROPs 3 [*Juglans regia*]	0.2238
	XP_018815447.1	14-3-3-like protein A [*Juglans regia*]	0.2179
	XP_018847629.1	bifunctional 3-dehydroquinate dehydratase/shikimate dehydrogenase, chloroplastic-like [*Juglans regia*]	0.2094
	XP_018821706.1	probably inactive leucine-rich repeat receptor-like protein kinase IMK2 [*Juglans regia*]	0.2087
	XP_018824157.1	rubisco accumulation factor 1, chloroplastic [*Juglans regia*]	0.2060
	XP_018824485.1	DNA replication licensing factor MCM6 [*Juglans regia*]	0.1852
	XP_018824515.1	40S ribosomal protein S4-1-like [*Juglans regia*]	0.1628
	XP_018852330.1	60S ribosomal protein L5 [*Juglans regia*]	0.1592
	XP_018812412.1	10 kDa chaperonin-like [*Juglans regia*]	0.1138
	XP_018833667.1	early nodulin-like protein 1 [*Juglans regia*]	0.1126
DEPs (S4/S2)	XP_018849923.1	exopolygalacturonase-like [*Juglans regia*]	53.2114
	XP_018834136.1	late embryogenesis abundant protein 2-like [*Juglans regia*]	48.7676
	XP_018834821.1	stress-induced protein KIN2-like [*Juglans regia*]	47.2469
	XP_018834135.1	late embryogenesis abundant protein 2-like [*Juglans regia*]	43.2973
	XP_018847735.1	xyloglucan endotransglucosylase/hydrolase protein 2-like [*Juglans regia*]	40.0863
	XP_018838067.1	late embryogenesis abundant protein 2-like [*Juglans regia*]	39.2749
	XP_018843119.1	mannan endo-1,4-beta-mannosidase 4-like [*Juglans regia*]	37.5167
	XP_018824181.1	ervatamin-B-like [*Juglans regia*]	37.0114
	XP_018848556.1	cysteine proteinase inhibitor B-like [*Juglans regia*]	36.5067
	XP_018814951.1	senescence-specific cysteine protease SAG39-like [*Juglans regia*]	35.1314
	XP_018818948.1	protein argonaute 4-like isoform X2 [*Juglans regia*]	0.1346
	XP_018840279.1	protein fluG [*Juglans regia*]	0.1293
	XP_018837465.1	uncharacterized protein At5g48480-like [*Juglans regia*]	0.1278
	XP_018837432.1	uncharacterized protein LOC109003650 [*Juglans regia*]	0.1175
	XP_018831451.1	metalloendoproteinase 2-MMP-like [*Juglans regia*]	0.1023
	XP_018847629.1	bifunctional 3-dehydroquinate dehydratase/shikimate dehydrogenase, chloroplastic-like [*Juglans regia*]	0.0802
	XP_018857107.1	60S ribosomal protein L11 [*Juglans regia*]	0.0728
	XP_018818184.1	aspartyl protease AED3-like [*Juglans regia*]	0.0630
	XP_018823302.1	osmotin-like protein isoform X1 [*Juglans regia*]	0.0625
	XP_018833667.1	early nodulin-like protein 1 [*Juglans regia*]	0.0384
DEPs (S4/S3)	XP_018845149.1	uncharacterized protein LOC109009208 [*Juglans regia*]	7.4335
	XP_018842192.1	aspartic proteinase nepenthesin-1-like, partial [*Juglans regia*]	7.2816
	XP_018819962.1	basic form of pathogenesis-related protein 1-like [*Juglans regia*]	7.1306
	XP_018824181.1	ervatamin-B-like [*Juglans regia*]	6.6394
	XP_018817239.1	serine carboxypeptidase-like 40 [*Juglans regia*]	6.5505
	XP_018845045.1	probable receptor-like protein kinase At5g24010 [*Juglans regia*]	6.4490
	XP_018807966.1	putative pectinesterase 63 [*Juglans regia*]	6.3742
	XP_018849923.1	exopolygalacturonase-like [*Juglans regia*]	6.2691
	XP_018860192.1	putative germin-like protein 2-1 [*Juglans regia*]	6.1890
	XP_018825423.1	protein MEN-8-like [*Juglans regia*]	6.0855
	XP_018839956.1	LRR receptor-like serine/threonine-protein kinase GSO1 [*Juglans regia*]	0.0909
	XP_018818184.1	aspartyl protease AED3-like [*Juglans regia*]	0.0896
	XP_018844051.1	probable glucan endo-1,3-beta-glucosidase A6 [*Juglans regia*]	0.0891
	XP_018851272.1	leucine-rich repeat receptor protein kinase EMS1-like [*Juglans regia*]	0.0837
	XP_018845266.1	cytochrome P450 704B1-like isoform X2 [*Juglans regia*]	0.0696
	XP_018848917.1	probable LRR receptor-like serine/threonine-protein kinase At4g36180 [*Juglans regia*]	0.0670
	XP_018841433.1	type III polyketide synthase B [*Juglans regia*]	0.0628
	XP_018814962.1	aldehyde oxidase GLOX-like [*Juglans regia*]	0.0609
	XP_018822513.1	putative leucine-rich repeat receptor-like serine/threonine-protein kinase At2g24130 [*Juglans regia*]	0.0558
	XP_018848853.1	fatty acyl-CoA reductase 2 [*Juglans regia*]	0.0480

## Data Availability

The data that support the findings of this study are available from the corresponding author upon reasonable request.

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
