# Peer review of "Comparative Proteomic Analysis of Floral Buds before and after Opening in Walnut (Juglans regia L.)"

_ijms, 2024, doi:10.3390/ijms25147878_

Round 1

Reviewer 1 Report

Comments and Suggestions for Authors

Walnut is an important economically tree species. And, heterodichogamy is the commonly characteristic of walnut trees. I have some questions and suggestions.

1. line 13-17, mainly described the characteristic of heterodichogamy, the main results are the bud formation and male and female differentiation. This could easily to mislead the readers and misunderstand the main purpose of the manuscript.

2. I could not understand the main purpose of the manuscript. the mechanism of dichogamy phenomenon of heterodichogamous? or the mechanism of bud differentiation?

3. Figure 3, not clear.

4. Firstly, the authors should to clear that, the flowering time of female and male is controlled by genetic or environmental impact.

5. I have no other suggestions, the main thing is, what is the significance of this study?

Author Response

Dear Reviewer,

Thank you for your valuable feedback on my article. My response to the question you provided is as follows.

  1. line 13-17, mainly described the characteristic of heterodichogamy, the main results are the bud formation and male and female differentiation. This could easily to mislead the readers and misunderstand the main purpose of the manuscript.

My answer: This is really a great suggestion! I have realized the problem and made targeted modifications. I deleted these statements “When the pistils mature and change into receptive before the stamens produce pollen, the condition is defined as protogyny, and when the pollen is desorbed before pistil receptivity, the condition is defined as protandry. Numerous genes related to flowering have been identified in related transcriptomics sequencing analysis, but proteomics shows less evidence for this mechanism of flowering response.” and added “The absence of female and male flowering periods seriously affects the pollination rate and fruit setting rate of walnuts, thereby affecting yield and quality. Therefore, the study of the characteristics and processes of flower bud differentiation helps to gain a deeper understanding of the regularity of the mechanism of heterodichogamy in walnuts.”

  1. I could not understand the main purpose of the manuscript. the mechanism of dichogamy phenomenon of heterodichogamous? or the mechanism of bud differentiation?

My answer: Based on our long-term observation of the growth of different walnuts varieties. Heterodichogamy is a stable genetic characteristic of walnuts. But in actual production practice, the differentiation of walnut flower buds has gone through multiple stages, with a time span of up to 10 months. We believe that selecting the period before and after the opening of male and female flower buds as the research object is more likely to discover some patterns. From the overall growth situation, flower bud differentiation exists at all stages and is also an important reason for this characteristic. So, we approach our work from this perspective.

  1. Figure 3, not clear.

My answer: Due to the insertion method of the image, we have made adjustments to determine its clarity.

  1. Firstly, the authors should to clear that, the flowering time of female and male is controlled by genetic or environmental impact.

My answer: Modifications have been made. Please refer to the revised paper for details.

  1. I have no other suggestions, the main thing is, what is the significance of this study?

My answer: We plan to start with proteins that directly affect flowering by using iTRAQ technology to search for differential expressed proteins before and after flowering. On this basis, we will continue to search for genes and pathways that may be related to metabolic activities such as flower bud differentiation and photosynthesis.

Thank you again for your suggestion.

Reviewer 2 Report

Comments and Suggestions for Authors

The aim of the work is to investigate, by using a proteomics approach, pistil and stamen walnut samples at the immature and mature stages. The study is well placed, in fact few works have used this approach for this topic.

However, some revisions are suggested.

                    The abstract is too long, please focus it on the main results achieved.

                    In the ‘Introduction’ section, please cite some other recent works about the examined topic, such as the one available at the link https://www.sciencedirect.com/science/article/abs/pii/S0304423821005781 , in which how the dichogamy type significantly influences female flower development rate in Juglans regia L. is discussed.

                    In the ‘Results' section, please improve the quality of figures 3 and 4, which are difficult to read.

Figure 6 also includes networks hard to read. I suggest to focus on sub-networks containing the most significant nodes and links.

Furthermore, Table 2 may be included in supplementary materials.

                    In the ‘Materials and Methods’ section, please specify the software tool used to create the PPI network

Furthermore:

                    A careful rereading of the entire manuscript is strongly recommended, in fact, the text contains numerous oversights and mistakes

Please check the reference format, especially about the year format. https://www.mdpi.com/journal/ijms/instructions

Comments on the Quality of English Language

Minor editing of English language required

Author Response

Dear Reviewer,

Thank you for your valuable feedback on my article. My response to the question you provided is as follows.

  • The abstract is too long, please focus it on the main results achieved.

My answer: Modifications have been made. Please refer to the revised paper for details.

  • In the ‘Introduction’ section, please cite some other recent works about the examined topic, such as the one available at the link https://www.sciencedirect.com/science/article/abs/pii/S0304423821005781, in which how the dichogamy type significantly influences female flower development rate in Juglans regia L. is discussed.

My answer: Thank you for your suggestion. We have cited this literature.

  • In the ‘Results' section, please improve the quality of figures 3 and 4, which are difficult to read.

My answer: Due to the insertion method of the image, we have made adjustments to determine its clarity.

Figure 6 also includes networks hard to read. I suggest to focus on sub-networks containing the most significant nodes and links.

My answer: Modifications have been made. Please refer to the revised paper for details.

Furthermore, Table 2 may be included in supplementary materials.

My answer: Thank you for your suggestion. We will make adjustments according to the editor's requirements.

  • In the ‘Materials and Methods’ section, please specify the software tool used to create the PPI network

My answer: Modifications have been made. Please refer to the revised paper for details.

Furthermore:

  • A careful rereading of the entire manuscript is strongly recommended, in fact, the text contains numerous oversights and mistakes

My answer: Modifications have been made. Please refer to the revised paper for details. In addition, we will seek formal companies to polish the article.

Thank you again for your suggestion.

Round 2

Reviewer 1 Report

Comments and Suggestions for Authors

The author has carefully revised the manuscript, and it is ready for acceptance.